# Exosomal Vimentin from Adipocyte Progenitors Protects Fibroblasts against Osmotic Stress and Inhibits Apoptosis to Enhance Wound Healing

**DOI:** 10.3390/ijms22094678

**Published:** 2021-04-28

**Authors:** Sepideh Parvanian, Hualian Zha, Dandan Su, Lifang Xi, Yaming Jiu, Hongbo Chen, John E. Eriksson, Fang Cheng

**Affiliations:** 1School of Pharmaceutical Sciences (Shenzhen), Sun Yat-sen University, Shenzhen 518107, China; sepideh.parvanian@abo.fi (S.P.); zhahlian@mail2.sysu.edu.cn (H.Z.); sudd6@mail2.sysu.edu.cn (D.S.); xilf@mail2.sysu.edu.cn (L.X.); chenhb7@mail.sysu.edu.cn (H.C.); 2Faculty of Science and Engineering, Åbo Akademi University & Turku Bioscience Centre, 20520 Turku, Finland; John.Eriksson@abo.fi; 3Key Laboratory of Molecular Virology and Immunology, Institut Pasteur of Shanghai, Chinese Academy of Sciences, Shanghai 200031, China; ymjiu@ips.ac.cn; 4Institute Pasteur of Shanghai and Institute of Pathogen Biology, University of Chinese Academy of Sciences, 52 Sanlihe Rd., Xicheng District, Beijing 100019, China

**Keywords:** mechanical stress, exosome, vimentin, wound healing, osmotic stress

## Abstract

Mechanical stress following injury regulates the quality and speed of wound healing. Improper mechanotransduction can lead to impaired wound healing and scar formation. Vimentin intermediate filaments control fibroblasts’ response to mechanical stress and lack of vimentin makes cells significantly vulnerable to environmental stress. We previously reported the involvement of exosomal vimentin in mediating wound healing. Here we performed in vitro and in vivo experiments to explore the effect of wide-type and vimentin knockout exosomes in accelerating wound healing under osmotic stress condition. Our results showed that osmotic stress increases the size and enhances the release of exosomes. Furthermore, our findings revealed that exosomal vimentin enhances wound healing by protecting fibroblasts against osmotic stress and inhibiting stress-induced apoptosis. These data suggest that exosomes could be considered either as a stress modifier to restore the osmotic balance or as a conveyer of stress to induce osmotic stress-driven conditions.

## 1. Introduction

Chronic wounds have become a significant source of major mortality and morbidity, which lead to high medical costs and poor quality of life [1]. Mechanical forces, such as compression, tension, shear stress, osmotic pressure, and gravity regulate the quality and speed of wound healing. After the injury, cells in the wound site encounter intensive changes in the mechanical forces which are induced by the injury itself or by the disruption of epithelial sheet force balance. Improper mechanotransduction can lead to impaired wound healing and scar formation [2,3,4].

Generally, when extracellular fluid osmolarity is higher than intracellular fluid, cells and tissues experience hyper-osmotic stress, and conversely, when intracellular solute concentrations exceed those outside the cell, hypo-osmotic stress happens [5]. Hypo-osmolarity induces cell swelling whereas hyper-osmolarity causes cell shrinkage. Such osmotic imbalances detrimentally affect water flux, cell volume, and signaling pathways involved in cell proliferation, cell migration, and apoptosis [6,7]. Kruse et al. reported that local hyperglycemia (hyper-osmotic stress), as well as systemic hyperglycemia, inhibited the migration of keratinocytes and fibroblasts as well as the re-epithelialization process [8]. Furthermore, hyper-osmotic stress modulates epidermal growth factor receptor transactivation [9] and inhibits the proliferation of human keratinocytes by increasing intracellular calcium levels [10]. One of the most dreaded consequences of the hyperglycemic crisis in diabetes is over-correction of hyperglycemia and hyper-osmolarity [11,12] which leads to excessive or prolonged edema, consequent cell swelling, and hindered wound healing [13,14].

Extracellular vesicles (EVs)—specifically, exosomes—secreted from stem cells have been shown to control inflammation, accelerate fibroblast migration and proliferation and prevent cell apoptosis [15,16]. The composition, biogenesis, and secretion of exosomes are strongly influenced by cellular stress conditions such as thermal and oxidative stress [17], radiation, photodynamic treatment, and chemotherapy [18], low pH condition [19], nutrient deficiency [20], anoxia, hypoxia [21] and cytoskeletal rearrangements [22]. Interestingly, exosomes also act as environmental stress modifiers by changing gene expression and phenotypic behaviors of the recipient cells. Thus, stress-induced changes in the composition of exosomal cargo are an efficient adaptive mechanism that helps cells to modulate intracellular stress conditions [23]. For example, exosomes from mesenchymal stem cells protect cells against oxidative stress and apoptosis [24,25]. Additionally, exosomes are considered as conveyers of the stress-mediated disease condition. For instance, exosomes from various cancer cells were shown to induce T-cell apoptosis and weaken adaptive immune cell function [26]. Additionally, enhanced immunosuppressive exosome release by thermal and oxidative stresses enhances the release of immunosuppressive exosomes and affects patients subjected to cytostatic and hyperthermal anti-cancer therapies adversely [17].

Cytoskeletal proteins act as a key component of the defense system against osmotic stress conditions whereas osmotic stress also regulates cytoskeletal protein rearrangement and expression. Specifically, vimentin—a major intermediate filament (IF) protein—plays an important role in the cell resistance to osmotic stress and protection against apoptosis [27]. Intuitively, vimentin by forming a cage-like network around the cell nucleus contributes to the mechanical integrity of the cell [28]. For example, astrocytes devoid of vimentin showed a less effective response to osmotic stress [29]. Our laboratory and others have validated the incorporation of vimentin into the exosomes from different cell lines [30,31,32,33,34]. Exosomal vimentin can attach to the cell surface at distinct sites via specific cell-surface receptors and initiate a cellular response [30,35]. Here, prompted by our previous findings underlying the crucial role of exosomal vimentin from adipocyte progenitors (APCs) in mediating wound healing, we aimed to investigate its contribution to osmotic mechanical stress during wound healing.

Additionally, the importance of cellular vimentin in wound healing and tissue repair has been the subject of several studies, the role of extracellular vimentin in these processes is not completely understood. We found that exosomes from wild type APCs promote wound healing in osmotic-stressed fibroblasts more than exosomes from vimentin knockout APCs. Our findings suggest that whereas wild type exosomes promote activities of osmotic-stressed fibroblasts, exosomes secreted from stressed adipocytes play an ominous role in wound progression and cell apoptosis. Importantly, exosomal vimentin plays an important role in the cell resistance to osmotic stress and cell protection against apoptosis which reflects the importance of vimentin and exosomes in mechanical cell behavior.

## 2. Results

### 2.1. WT-APCs Tolerate Osmotic Stress Better than Vim−/−APCs

We first optimized the osmotic stress conditions, and 150 mOsm and 200 mM sorbitol were selected for hypo-osmotic and hyper-osmotic stress conditions, respectively for cell morphology and viability assays (Appendix A). We then cultured both wild type (WT) and vimentin knockout (Vim−/−) APCs under hypo (H−) and hyper-osmotic (H+) stress conditions (Figure 1a). Since osmotic stress can affect cell volume, we measured these changes in the cell volume along a particular axis as a percentage of its original length and we represented it here as cell elongation. The results showed that there is an increase in cell elongation for both WT (11%) and Vim−/−APCs (21%) in hypo-osmotic stress media, whereas there was a reduction in hyper-osmotic stress media (33% for WT and 51% for Vim−/−APCs) (Figure 1d). Additionally, the confluency percentage -the percentage of the surface that is covered by the cells- was decreased after 24 h for WT and Vim−/−APCs in hypo- (40% and 28%) and hyper-osmotic media (32% and 21%) (Figure 2b). The same reduction was observed for WT and Vim−/−APCs number after 24 h incubation in hypo- (67% and 72%) and hyper-osmotic media (67% and 75%) (Figure 2c). These results show that WT-APCs can tolerate osmotic stress better than Vim−/−APCs.

### 2.2. Osmotic Stress Significantly Increases the Exosome Secretion by WT and Vim−/−APCs

We first isolated exosomes from cell culture supernatants produced by an equal amount of WT and Vim−/−APC cells from normal, hypo, and hyper-osmotic stress conditions by differential centrifugation and ultracentrifugation and then evaluated by transmission electron microscopy (TEM), Nanoparticle Tracking Analysis (NTA), and Western blot. Exosomes with diameters ranging between 30–300 nm (Figure 2a,b,d). The results from Immuno-EM images confirmed the existence of vimentin and CD9 in isolated exosomes. Although the vimentin attachment site to the exosomes is not obvious from the immune-EM images, according to our previous results from STED microscopy, vimentin could exist inside (intravesicular) or attached to the surface of the exosomes (vesicular surface). Additionally, the structure and the organization of exosomal vimentin remained to be further explored. Additionally, the expression of exosomal markers Hsp70, CD9, CD63, and CD81 was confirmed by Western blot analysis (Figure 1c). Then, we investigated whether osmotic stress affects the size and the number of exosomes secreted by WT and Vim−/−APCs. According to DLS analysis, in normal condition, there was a 17% reduction in the average size of Vim−/−Exos compared to WT-Exos. Osmotic stress increased the size of secreted exosomes from both WT and Vim−/−APCs, while this increase was higher for Vim−/−Exos (57% in hyper-osmotic and 74% in hypo-osmotic stress) compared to WT-Exos (17% in hyper-osmotic and 52% in hypo-osmotic stress) (Figure 2e,f). In the next step, we measured the quantity of secreted exosomes by BCA protein assay, fluorescence intensity analysis, and densitometry of Western blots. The results showed a clear tendency of increased exosome quantity by both hypo and hyper-osmotic stress in all three types of measurements (Figure 2g–k, *n* = 5). For instance, CD9 expression in Vim−/−Exos was upregulated by 4-fold by hypo- and 6-fold increase under hyper-osmotic stress, which was a 7.5-fold increase by hypo- and 8-fold increase by hyper-osmotic stress for WT-Exos (Figure 2g,h). Interestingly, there were no significant changes in the concentration of Vim−/−Exos compared to WT-Exos. According to these results, we conclude that osmotic stress can up-regulate exosome secretion, and WT-APCs are more susceptible to stress-mediated up-regulation of exosome secretion compared to Vim−/−APCs. For future clarification, we used following abbreviations for different isolated exosomes: exosomes from wild type adipocytes (WT-Exo), exosomes from vimentin knockout adipocytes (Vim−/−Exo), exosomes from wild type adipocytes under hypo-osmotic stress (WT-H-Exo), exosomes from vimentin knockout adipocytes under hypo-osmotic stress (Vim−/−H-Exo), exosomes from wild type adipocytes under hyper-osmotic stress (WT-H + Exo) and exosomes from vimentin knockout adipocytes under hyper-osmotic stress (Vim−/−H + Exo).

### 2.3. WT-Exos Promotes Proliferation and Prevents Apoptosis of Osmotic-Stressed HDFs

We first confirmed the efficiency of APC-Exos uptake by performing in vitro uptake assay. The microscopy results revealed that APC-Exos was already significantly taken up by HDF recipient cells on 6 h and peaked at 24 h (Figure 3a,b). Then we performed proliferation (confluency) and apoptosis assays to evaluate the effect of APC-Exos on osmotic-stress HDFs. HDFs were incubated in hypo and hyper-osmotic media and co-cultured with WT-Exos and Vim−/−Exos. The confluency rate of osmotic-stressed HDFs was increased significantly only after WT-Exos treatment (81% for hypo and 68% hyper-osmotic, *p* < 0.001) (Figure 3c). Furthermore, caspase 3/7 green apoptosis staining assay revealed that the co-culture of osmotic-stressed HDFs with WT-Exos significantly suppressed both hypo and hyper osmotic-induced apoptosis compared to non-treated osmotic-stressed HDFs (90% for hypo and 92% for hyper-osmotic stress, *p* < 0.001). The apoptosis rates were similar among the non-treated normal HDFs (control) and both hypo and hyper-osmotic-stressed HDFs co-cultures with WT-Exos (Figure 4a,b). Poly(ADP-ribose) polymerase-1 (PARP-1) is a nuclear enzyme that is involved in cellular response to DNA damage. Once PARP is cleaved by caspase during apoptosis, its DNA repair function is impaired. Here, we assessed the expression levels of PARP and its cleaved form, and there was no change in the total expression of PARP among control and both hypo and hyper-osmotic-stressed HDFs co-cultures with WT-Exos (Figure 4e–g). Thus WT-APC-Exos modify osmotic stress-induced apoptosis in fibroblasts.

### 2.4. Osmotic-Stress Induced Exosomes Influence HDFs Proliferation and Apoptosis

In the parallel experiments, HDFs co-cultured with WT-H- Exos, Vim−/− H- Exos, WT-H+ Exos, and Vim−/− H+ Exos were evaluated for proliferation and apoptosis rates. The cell proliferation was slowed down by hypo and hyper osmotic-stressed exosomes from both WT (69% and 73%) and Vim−/−APCs (85% and 79%), respectively (Figure 3d). Furthermore, the number of apoptotic HDFs was significantly increased after treatment with exosome from osmotic-stressed APCs compared with control (*p* < 0.001). There was no significant difference in apoptosis rate between osmotic-stressed HDFs and normal HDFs following treatment with Vim−/− H− Exos, WT-H+ Exos, and Vim−/− H+ Exos (Figure 4c,d). Furthermore, Western blot analysis showed that cleaved PARP was significantly increased by 2 fold after normal HDFs treatment with WT-H- Exos, Vim−/− H− Exos, WT-H+ Exos, and Vim−/− H+ Exos (Figure 4e,h,i). Taken together, these results indicated that WT-Exos significantly suppressed both hypo and hyper osmotic-induced apoptosis, while exosomes from osmotic-stressed APCs can induce apoptosis.

### 2.5. WT-APC-Exos Affect Collagen Fiber Orientation and Promote ECM Production by Osmotic Stressed HDFs

Collagen architecture and orientation are the major determinants of the fibroblasts’ mechanical behavior to meet mechanical stress. To study the collagen fiber orientation, we used CDMs secreted from osmotic-stressed and normal HDFs, where HDFs were treated with WT-Exos or Vim−/−Exos. Since WT-Exos promoted osmotic-stressed-HDFs migration and proliferation, we hypothesized that orientation changes in the collagen fibers in CDMs are due to the orientation changes in the originated cell. Representative images of native cells (Figure 5a) and their corresponding orientation histograms (Figure 5b) showed a random cell orientation for non-treated HDFs (controls) and treated HDFs with Vim−/−Exos, while WT-Exos treated HDFs were aligned in the certain direction. Interestingly, there was a preferred fiber direction in HDFs treated with WT-Exos and fiber alignment was much more homogenous in this group compared to non-treated HDFs and HDFs treated with Vim−/−Exos (Figure 5c,d). Figure 5b,d show the distribution of the fibers’ orientation while dominant orientation is pointed out with the red arrows. According to the color-coded bar in Figure 5f, fibers in WT-Exos samples appear in the same color (same direction) while no preferred fiber alignment was observed in the controls and Vim−/−Exos samples. In line with these results, as shown in Figure 5e, there were stronger fluorescent signals of collagen I in CDMs treated with WT-Exos, indicating that collagen I deposition was enhanced in the presence of WT-APC-Exos. In short, these results showed that the directionality of collagen fibers is similar to the directionality of the original cells, and compared to other treatments, WT-Exos can guide cells mainly to be oriented at the same angle.

### 2.6. WT-APCs-Exos Promote Wound Healing In Vivo

In cellular models relevant to wound healing, we found that WT-APCs tolerate osmotic stress better than Vim−/−APCs, and WT-APC-Exos promote osmotic stressed HDFs proliferation. Thus, we questioned whether exosomal vimentin is also an important determinant in wound healing in in vivo osmotic-stressed mouse model. To this aim, we constructed hypo-osmotic and hyper-osmotic stress models upon a full-thickness excisional injury to the dorsal skin of mice and then treated the wounds with exosomes for 5 days. Results demonstrated that the wounds treated with WT-APC-Exos healed faster than mice treated with Vim−/−APC-Exos. Furthermore, WT-APC-Exos treated wounds healed almost completely with a minimum scar size, while the epithelial layer of wounds from the Vim−/−APC-Exos and control groups still had obvious scars on day 10 (Figure 6a,b). Histological analysis of the wound tissue revealed that WT-APC-Exos significantly reduced inflammation and immune cell infiltration when compared to Vim−/−APC-Exos and control (Figure 6c). Moreover, RT-qPCR analysis shows that IL-12 was significantly higher in the Vim−/−APC-Exos group or control group when compared to the WT-APC-Exos group (Figure 6d), suggesting a protective role of exosomal vimentin against inflammation.

## 3. Discussion

In the present study, prompted by our previous findings underlying the involvement of exosomal vimentin in promoting wound healing, we explored that exosomal vimentin may have a role in cell resistance to osmotic stress and cell protection against apoptosis.

Osmotic stress by creating cell volume perturbations can affect InsR, ERK- and FAK signaling pathways which can cause changes in cell characteristics such as viability and proliferation [36,37]. Cytoskeletal cross-linking by forming a sponge-like interior bonded to the membrane at the periphery helps to distribute osmotic stress throughout the cell volume [38]. Here we showed that WT-APCs tolerate osmotic stress better than Vim−/−APCs [27,28] and lack of vimentin makes cells significantly vulnerable to environmental stress which indicates the importance of vimentin in cell resistance to osmotic stress and apoptosis.

Exosome cargo contains protein and RNA molecules that are not randomly loaded into the exosomes and the composition of exosomes is remarkably influenced by environmental challenges that define the outcome of communication between the exosome-producer and the recipient cell [23]. Here, in a series of parallel experiments, we showed that osmotic stress could cause the enhanced release of APCs-derived exosomes and increased their size. This might partly explain the initially proposed role of exosomes as an alternative way of eliminating waste products such as waste membranes, harmful RNA, or proteins to maintain cellular homeostasis. Furthermore, the exosome containing cell waste material can likely affect neighboring cells. In this manner, increasing exosome release could be a way of communication with neighboring cells about intracellular stress and possibly induce the same pathological condition [39]. This result ties well with previous studies wherein there is a link between autophagy and exosome release, describing exosome as an alternative route to dispose of cellular waste when the transport through the degradative or lysosomal pathway is obstructed due to stress [40].

Besides the role of exosomes as cellular waste disposal compartments, more importantly, exosomes can act as signal carriers of signaling, toxic, and regulatory molecules to modify other cells’ function in normal and disease conditions [41]. According to our results, WT-APCs-Exo could be taken up and promote wound healing of osmotic-stressed HDFs remarkably by affecting their migration, proliferation, and ECM production suggesting that HDFs employ exosome-mediated cell communication to manage cellular stress conditions.

As we discussed earlier, cellular stress conditions can reflect in the protein and RNA content of exosomes [42]. Here, we hypothesized that, osmotic stressed APCs-Exos could mediate the communication of stress-related signals and that the content of these exosomes could induce the cellular stress in the recipient HDF cells. We observed profound differences in the function of exosomes derived from normal and osmotic-stressed APCs. Accordingly, WT-Exos were involved in stress resistance and promoting cell proliferation, cell directional migration, ECM production, and apoptosis inhibition whereas H + Exos and H-Exos were associated with a reduction in cell confluency and apoptosis progression. This result could indicate exosomes either as stress modifiers in carrying molecules from cells of origin to the peripheral circulation to restore the osmotic balance or as a conveyer of stress to induce osmotic stress-driven conditions.

It has been shown that exosomes from cells grown under stress conditions induce cytoskeletal rearrangements and extracellular matrix remodeling [43]. Vimentin is a highly stable, stress-resistant cytoskeleton protein which could be released into the extracellular space as an extracellular protein and bind to the cell-surface of repair-modulating cells on the injury. The release of incorporated vimentin into the exosomes which represent vimentin-containing vesicles is one of the potential sources of extracellular vimentin pool [31,35]. Furthermore, cytoskeletal blocking including vimentin considered as cellular stress could cause the enhanced release of exosomes and might change the protein and RNA content of exosomes [22]. Our results showed that whereas exosomes from WT-APCs play an important role in cell resistance to osmotic stress and cell protection against apoptosis, the exosomes secreted from Vim−/−APCs may play a more ominous role in this process. Furthermore, we observed that WT-APCs are more susceptible to stress-mediated up-regulation of exosome secretion compared to Vim−/−APCs. These observations may imply the importance of vimentin in mechanical cell behavior as a critical regulator of wound healing.

Additionally, Vimentin has been shown to promote fibroblasts’ motility, directionality, and ability to organize ECM proteins [44,45]. We previously reported that unlike Vim−/−Exos, WT-Exos can drive directional cell motility optimally in a certain direction towards the wound area. Our results from the current study go beyond previous reports, showing that despite an increase in ECM production by both WT and Vim−/−Exos, there is a preferred alignment direction of collagen I fibers under treatment of WT-Exos, whereas, by Vim−/−Exos treatments, no apparent alignment behavior can be detected. Such a high density aligned ECM network may indicate HDFs as direction-selective cells and exosomal vimentin as a stimulator for this response.

## 4. Materials and Methods

### 4.1. Cell Culture and Exosome Isolation

Cell Culture: Adult-HDFs (Human Dermal Fibroblast) were purchased from ScienCell, Carlsbad, CA, USA and mouse APCs were purchased from ZenBio, Inc., Durham, NC, USA. Vim−/−3t3l1 was obtained from Turku Bioscience Centre. Both HDFs and 3t3l1 cells were cultured in Dulbecco’s Modified Eagle Medium (DMEM) (Lonza, Switzerland) supplemented with 1% l-glutamine (Lonza, Switzerland), 0.5% penicillin/ streptomycin (Gibco Life Technologies Ltd., New York, NY, USA) and10% fetal bovine serum (FBS) (Thermo Fisher Scientific, Waltham, MA, USA).

Osmotic stress induction: All the experiments were performed in both hyper and hypo-osmotic stress conditions. For hypo-osmotic stress, cells were treated with normal growth media diluted appropriately in deionized water for the following ratios: 1:9, 1:4, and 1:1 to obtain 30 mOsm, 60 mOsm, and 150 mOsm, respectively. For hyper-osmotic stress, cells were treated with normal growth media supplemented with 100 mM NaCl, 200 mM sorbitol, or 200 mM glucose for an increase of 200 mOsm kg^−1^ H_2_O. Considering the cell viability and morphology results, 150 mOsm for hypo-osmotic and 200 mM sorbitol for hyper-osmotic stress were selected as optimal conditions. Cell number and cell confluency assays were quantified using the IncuCyte S3™ instrument. Cell elongation was measured using ImageJ software.

Exosome isolation: Exosomes were isolated from five conditioned media: hypo-osmotic stressed WT-APCs (WT-H-Exo), hypo-osmotic stressed Vim−/−APC (Vim−/− H − Exo), hyper-osmotic stressed WT-APCs (WT-H + Exo), hyper-osmotic stressed Vim−/−APC (Vim−/− H + Exo), and normal APC (WT-APCs: control) as described before.

For isolating exosome, at 70% confluency, cells were washed with PBS, and growth media supplemented with 0.5% Exosome-Depleted fetal bovine serum (FBS) (Thermo Scientific, USA) was replaced. All the cell culture protocols were carried out at 37 °C in a humidified 5% CO_2_ environment. After 24 h, culture supernatant was collected, and exosomes were isolated using a differential centrifugation protocol. Collected conditioned media were centrifuged at 300× *g* for 10 min to remove cellular debris. The supernatant then was transferred to a new 15 mL conical tube and centrifuged at 2000× *g* for 20 min to isolate apoptotic bodies. This was followed by transferring the supernatant to a sterile Ultra-Clear tube (Beckman Coulter, Sharon Hill, PA, USA) and centrifugation in a Beckman Coulter Optima™ L-80XP Ultracentrifuge to isolate microvesicles by 40 min centrifugation at 10,000× *g*. After this, the supernatant was again collected and centrifuged at 100,000× *g* avg for 90 min to pellet exosomes. The resulting exosome pellet was resuspended in 1× PBS and stored at −80 °C for future use. All procedures were performed at 4 °C.

### 4.2. Exosome Characterization and Analysis

Before the use of exosomes in further experiments, isolated exosomes were evaluated for morphology by transmission electron microscopy (TEM), particle concentration, distribution, and size by a dynamic light scattering (DLS) and nanoparticle tracking analysis (NTA) and exosomal markers by Western blot.

Transmission electron microscopy (TEM): TEM of isolated exosomes was performed by the EV Core at the University of Helsinki. Exosomes were loaded on carbon-coated and glow discharged 200 mesh copper grids with pioloform support membrane. Samples were then fixed with 2.0% PFA in NaPO_4_ buffer and stained with 2% neutral uranyl acetate with embedding in uranyl acetate and methylcellulose mixture (1.8/0.4%). For immunostaining, samples were blocked with 0.5% BSA in 0.1 M NaPO_4_ buffer (pH 7.0), incubated with anti-CD9 (1:50 dilution, Novus biological) or anti-vimentin (1:200 dilution, Novus biological) in 0.1% BSA/NaPO_4_ buffer. Then, samples were incubated with 15 nm goat anti-rabbit IgG (1:80 dilution, BBI Solutions, Cardiff, UK) in 0.1% BSA/NaPO_4_ buffer, washed with the NaPO_4_ buffer, and deionized water and negatively stained. Exosomes were viewed with transmission EM using Jeol JEM-1400 (Jeol Ltd., Tokyo, Japan) operating at 80 kV. Images were taken with Gatan Orius SC 1000B CCD-camera (Gatan Inc., Pleasanton, CA, USA) with 4008 × 2672 px image size and no binning.

Western blot analysis: Exosomes and cell lysate samples were lysed in RIPA buffer (5 mM EDTA, 150 mM NaCl, 1% NP40, 1% sodium deoxycholate, 1% SDS 20% solution, 50 mM Tris-HCl, pH 7.4) containing protease/phosphatase inhibitor cocktail (Cell Signaling, Danvers, MA, USA), heated to 95 °C for 5 min and subsequently cooled on ice. The samples were measured for total protein concentration using a Pierce BCA protein assay kit (Thermo Scientific Pierce, Rockford, IL, USA) and were analyzed with a spectrophotometer at 562nm (Hidex Plate Reader, Turku, Finland). Samples (30 μg of protein per well) were separated on one-dimensional sodium dodecyl sulfate-polyacrylamide gel electrophoresis (SDS-PAGE) 12% gel. Proteins were transferred to PVDF membrane (BioRad Laboratories, Hercules, CA, USA), blocked in 5% non-fat powdered milk in TBS-T (0.5% Tween-20), and probed with exosomal characteristic markers antibodies, CD9 (1:500), CD63 (1:1000), CD81 (1:1000), Hsp70(1:1000), GAPDH (1:500) (all from System Biosciences, Palo Alto, CA, USA) and vimentin (1:500, Biolegend, San Diego, CA, USA) overnight. Membranes were washed three times, 5 min in TBS-T to rinse off the residual primary antibodies and probed using their respective secondary antibodies at a 1:10,000 dilution. The signals were visualized by the ECL Prime Western Blotting Detection Reagent (Advansta, San Jose, CA, USA) and iBright CL1500 western blot imaging system (Thermo Fisher Scientific, Waltham, MA, USA).

Nanoparticle Tracking Analysis (NTA): Particle number and size distribution of the exosome were performed by the EV Core at the University of Helsinki. Exosome samples were diluted with PBS and measured by NTA instrument LM14C (NanoSight LTD., London, UK) equipped with blue (404 nm, 70 mW) laser and sCMOS camera. Data were analyzed with Nanosight software v3.0, using threshold 5 and gain 10.

Dynamic Light Scattering (DLS): The size/diameter of the isolated exosomes were measured by DLS (Malvern Panalytical, Malvern, UK). Samples (100 μL) were diluted to 1000 μL in PBS, added to a cuvette. After removing air bubbles, size, and density for each sample were measured (*n* = 3).

### 4.3. Exosome Labeling and Quantification

Purified exosomes were fluorescently labeled with 1,1′-Dioctadecyl-3,3,3′,3′-tetramethylindocarbocyanine perchlorate (Dil) dye (Thermofisher), a fluorescent dye that is incorporated into the cell membrane for 1 h at 37 °C, and excess dye was removed by ultracentrifugation at 100,000× *g* for 2 h. Exosomes were then re-suspended in PBS at the concentration of ~1 μg protein/μL. Fluorescent intensity (Ex 530, Em 590) was measured for the equal amount of the labeled exosomes from each sample (Hidex Plate Reader, Finland) and values were then corrected for differences in the total number of the viable cells for each condition.

### 4.4. HDFs Proliferation Assay

HDF cells were seeded with an initial density of 5 × 104 cells/well in a 24 well plate. After 24 h, the media was replaced with hypo-osmotic and hyper-osmotic media. After 4 h, osmotic-stressed HDFs were treated with WT-Exos (100 µg/mL) and Vim−/−Exos (100 µg/mL), while normal HDFs were treated with the same amount of WT-H-Exo, Vim−/− H-Exo, WT-H + Exo, and Vim−/− H + Exo. Normal and osmotic-stressed HDFs were used as controls. The effect of the exosomes on the proliferation of HDFs was measured as cell confluency percentage using the IncuCyte S3™ instrument. Three images were taken per well every 2 h for 48 h and images were analyzed using IncuCyte S3 software. All proliferation assays were performed in triplicates.

### 4.5. Exosome Uptake by HDFs

To determine whether APCs-Exos can be efficiently taken up by HDFs, we performed an exosome uptake in vitro experiment. DiI-labeled APCs-Exos were incubated with HDFs at 70% confluency for 48 h, followed by fixation with 4% paraformaldehyde, and then stained for nuclei using 0.3 µg/mL DAPI (Thermofisher). Cells were imaged with a 63× objective (Immersion: Oil, Numerical Aperture: 1.4) under a 3i CSU-W1 spinning disk confocal microscope (Intelligent-imaging, Denver, CO, USA) equipped with a camera (Photometrics, Tucson, AZ, USA).

### 4.6. Cell Apoptosis and Analysis

HDF cells were seeded at 5 × 104 cells/well in a 24 well plate. After 24 h, the media was replaced with hypo-osmotic and hyper-osmotic media for 4 h. Osmotic-stressed HDFs were treated with 100 µg/mL of WT-Exos and Vim−/−Exos while normal HDFs were treated with 100 µg/mL of WT-H-Exo, Vim−/−H-Exo, WT-H + Exo, or Vim−/−H + Exo. Normal and osmotic-stressed HDFs were used as controls. The effect of the APC-Exos on HDFs apoptosis was determined using IncuCyte^®^ caspase-3/7 green apoptosis assay reagent (Sartorius, Bohemia, NY, USA). Three images were taken per well every 2 h for 48 h by IncuCyte™ S3. The apoptosis rate was quantified by acquired fluorescent signal using an integrated object counting algorithm by IncuCyte™ S3 software and then normalized against cell confluency. All apoptosis assays were performed in triplicates. Western blot analyses were performed as described previously with the β-actin (Cell Signaling Technology, Danvers, MA, USA) and PARP-1 (Santa Cruz Biotechnology, Santa Cruz, CA, USA) primary antibodies and the corresponding secondary antibodies. Signals were quantified using ImageJ software.

### 4.7. ECM Staining and Quantification

For ECM production experiments, cell-derived matrices (CDM) secreted by HDFs were used to mimic the three-dimensional (3D) nature of in vivo microenvironments. Briefly, sterile coverslips in 24 well plates (Greiner Bio-One GmbH, Kremsmünster, Austria) were coated with 0.2% pre-warmed gelatin (Sigma-Aldrich, St. Louis, MO, USA) for 1 h at +37 °C and cross-linked using 1% (*v*/*v*) glutaraldehyde (Sigma-Aldrich, St. Louis, MO, USA) for 30 min at room temperature. Coverslips then were treated with 1 M glycine (Sigma-Aldrich) for 20 min at room temperature. Prepared coverslips were incubated with 5 × 104 HDFs to form a confluent monolayer. To make CDMs from osmotic-stressed HDFs, after 24 h, growth media were replaced with osmotically stressed media, and cells were treated with WT-Exos or Vim−/−Exos (100 µg/mL). Coverslips were then treated with 50 μg/mL sterile ascorbic acid (Sigma-Aldrich) every other day for 10 days. Fibroblasts were extracted from CDMs by adding a prewarmed sterile-extraction solution (for 50 mL buffer: 1 mL of NH_4_OH, 250 μL of Triton X-100, and 48.75 mL of PBS). CDMs were treated with 10 μg/mL of DNase I (Roche, Basel, Switzerland) and washed with PBS. CDMs from normal and osmotic-stressed HDFs were used as the controls.

CDMs architecture was analyzed by immunofluorescence staining of CDMs using collagen I (Novusbio, Littleton, CO, USA; 1:200) antibody and visualized using a 3i spinning disk confocal microscope (20× objective, numerical Aperture: 0.8) equipped with an Orca Flash 4 v2 C11440-22CU Scientific CMOS camera (Hamamatsu, Ammersee, Germany). Collagen fluorescent intensity was measured by ImageJ software. Three samples of each condition measured for the mean intensity (*n* = 10). Subsequently, the values from individual immunofluorescence images with maximum intensity projection were used for the statistical analysis by using a paired *t*-test; *p* < 0.05 was considered significant.

To characterize the orientation of collagen fibers, OrientationJ, a series of ImageJ plugins was used. The local orientation of fibers showed as a color image with the orientation being encoded in the color. The fibers with the same orientation appeared in the same color. The orientation of the fibers was quantified for every pixel of the image based on the structure tensor as a histogram expressing the fiber distribution of orientation.

### 4.8. Mouse Skin Injury Model and Treatment

SPF Balb/c mice (male, 5–8 weeks old) were purchased from the laboratory animal center of Sun Yat-Sen University. Animal procedures were approved by the Institutional Animal Care and Use Committee (IACUC) at Sun Yat-Sen University (approval number: SYSU-IACUC-2019-00001). Mice were fed with irradiated maintenance fodder and sterile water. Mouse skin excisional wound model including the hypo-osmotic stress model and hyper-osmotic stress model was constructed according to previous reports [46]. To access wound healing, the dorsal dermal hair was removed by using the depilatory cream (VEET). Mice were injected with anesthetic by intraperitoneal injection, and then the mice went into a coma. The electric hairdresser (AUX-A8) was firstly used to remove part of the mice’ s dermal hair, after that, the depilatory cream (VEET) was evenly applied to the surgical site in a thin layer for 5 min, and then cotton balls which dipped in distilled water were used to gently wiped on the depilatory cream to remove both depilatory cream and the dorsal dermal hair. Then the bare skin was treated with a hypertonic or hypotonic solution-soaked gauze for 1 min. Sorbitol (200 mM solution) was used as a hypertonic solution for the hyper-osmotic stress model and distilled water was used as a hypotonic solution for the hypo-osmotic stress model. Treatments were repeated daily at a specific time for 7 days. Then, mice were anesthetized by intraperitoneal injection of 1% pentobarbital sodium (Sigma-Aldrich) at a dose of 10 mL/kg. A 10 mm diameter circle full-thickness skin wound was created on the midline of the mice’s back spine by surgery. Mice were randomly divided into six treatment groups: Hypo-osmotic stress (H−) and hyper-osmotic stress (H+) as controls, H- treated with WT-APCs-Exos (H-WT-Exo), H- treated with Vim−/−APC-Exos (H-Vim−/−Exo), H+ treated with WT-APCs-Exos (H + WT-Exo) and H+ treated with Vim−/−APC-Exo(H + Vim−/−Exo). After three days of wounding, each group was given an intraperitoneal injection of related APC-Exos (160 μg/μL, 5 mg/kg) once a day for five days. Control groups did not receive any treatments. Weight and wound diameter was recorded and was analyzed with ImageJ. The wound tissues and spleen were collected on day 10 and each tissue was divided into two parts, one part stored at −80 ℃, and the other was submerged in 4% paraformaldehyde for further analysis.

### 4.9. Histology

At certain intervals, mice were sacrificed, and their wound tissue samples were carefully biopsied and fixed in a 4% paraformaldehyde solution. After 48 h of fixation, tissue samples were dehydrated in a graded series of ethanol concentrations and embedded in paraffin. Tissue samples were cut into 5 μm slices and stained with hematoxylin and eosin (H&E). Histological changes were visualized (Nikon Eclipse Ti2, USA) and recorded.

### 4.10. RNA Isolation and qPCR Analysis

Total RNA from skin tissue and spleen was isolated by TRIZOL reagent (TaKaRa, Tokyo, Japan). Samples were ground down in the mortar before adding TRIZOL reagent. NANODROP ONE (Thermo Fisher Scientific, Waltham, MA, USA) was used to measure RNA concentrations. RNA was reversely transcribed into complementary DNA (cDNA) by using HiScript III RT SuperMix for qPCR (+gDNA wiper) (Vazyme, Jiangsu, China) with T100TM Thermal Cycler (BIO-RAD, Dalkeith, UK), and then quantified by qPCR using 2x SYBR Green qPCR Mix (Sigma-Aldrich, St. Louis, MO, USA) with LightCycler^®^ 96 (Roche, Basel, Switzerland). All processes were following the manufacturer’s instructions. Relative gene expression folding changes were identified with the 2−ΔΔCt method. The primers used in this study are summarized in Table 1.

### 4.11. Statistics

Statistical analyses were performed using Student’s *t*-test or ANOVA as appropriate in SPSS software and the results were considered significant when *p* < 0.05.

## 5. Conclusions

Altogether, the results of this study for the first time indicated that exosomes can be considered as a complex information package to either modify and restore the osmotic balance or to convey and induce osmotic stress-driven condition, while exosomal vimentin significantly contributed to this process. However, more mechanistic studies are needed to illuminate how stress conditions affect exosome-mediated intercellular communication, signaling pathways, and phenotypic behavior of recipient cells. Furthermore, advances in exosome isolation and purification techniques may help to study the exosomal vimentin filament organization and characterization during stress conditions. Such studies could significantly broaden our understanding of exosomes as novel cell-free agents in modifying cellular stress.

## Figures and Tables

**Figure 1 ijms-22-04678-f001:**
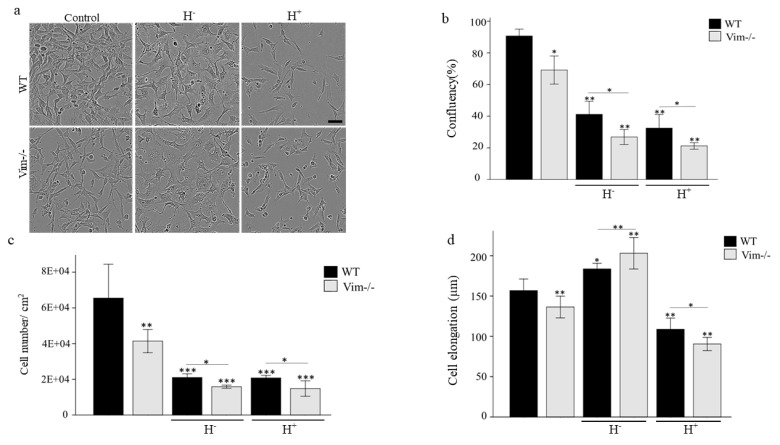
WT-APCs tolerate osmotic stress better than Vim−/−APCs. (**a**) Representative images of WT and Vim−/−APCs incubated in hypertonic (+ 200 mOsm) and hypotonic (150 mOsm) after 24 h of treatment. Scale bar: 100 μm. (**b**) Confluency percentage (**c**) Cell number (**d**) Elongation measurements of WT and Vim−/−APCs after 24 h of treatment using an IncuCyte S3 system. Normal media (300 mOsm) were used as control. * *p* < 0.05, ** *p* < 0.01, *** *p* < 0.00.

**Figure 2 ijms-22-04678-f002:**
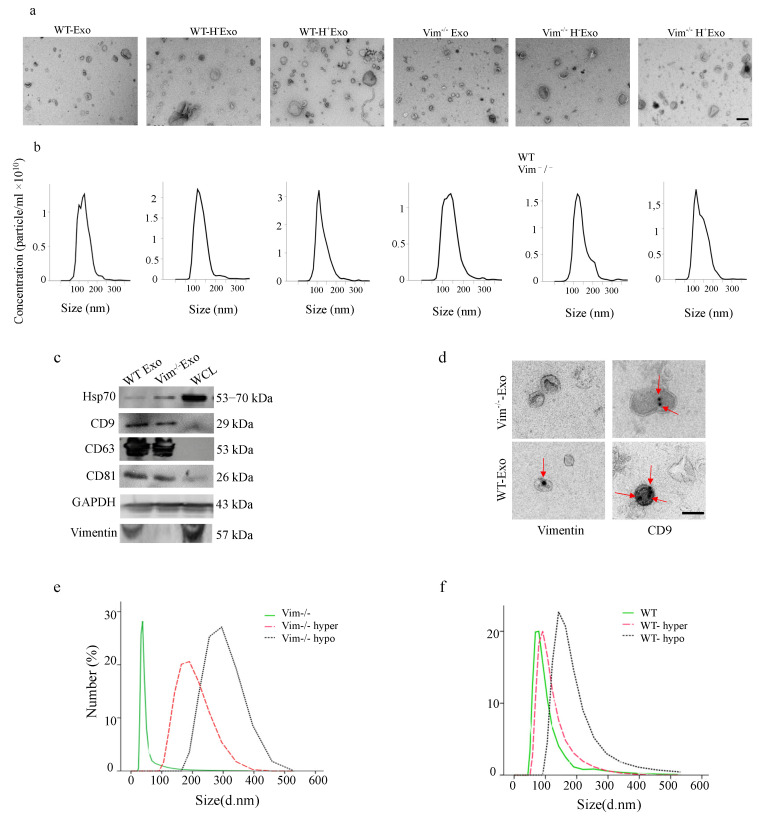
Osmotic stress increase exosome release by WT and Vim−/−APCs. (**a**–**d**) Characterization of exosomes derived from WT and Vim−/−APCs. (**a**) Representative images of transmission electron microscopy of isolated WT-Exos and Vim−/−Exos. Scale bar = 200 nm. (**b**) The size distribution of WT-Exos and Vim−/−Exos measured by Nanoparticle Tracking Analysis (NTA). (**c**) Detection of CD81, CD9, CD63, Hsp70, GAPDH, and vimentin expression in exosomes by Western blotting. WCL is the whole cell lysate from WT-APC as a control. (**d**) Immuno labeling of WT-Exos and Vim−/−Exos with vimentin and CD9 antibody. Red arrows show detected vimentin and CD9. Scale bar = 100 nm. (**e**) Dynamic light scattering size distribution analysis of exosomes released from Vim−/− and (**f**) WT-APCs after 24 h of treatment. (**g**) Western blot and (**h**) Pixel density quantification of CD9 antibody. (**i**) Fluorescence microscopy and (**j**) quantification analysis of DiI-labeled exosomes (per million cells) from APCs conditioned medium after 24 h incubation in hypertonic (+200 mOsm) and hypotonic (150 mOsm) conditions. Quantification was based on the red fluorescent intensity of Dil-labeled exosome. Scale bar = 50 μm. (**k**) Exosome concentration from WT and Vim−/−APCs after 24 h of treatment by BCA kit. Normal media (300 mOsm) were used as control. Exosomes from WT and Vim−/−APCs incubated in normal media (300mOsm) used as control. ** *p* < 0.01, *** *p* < 0.001.

**Figure 3 ijms-22-04678-f003:**
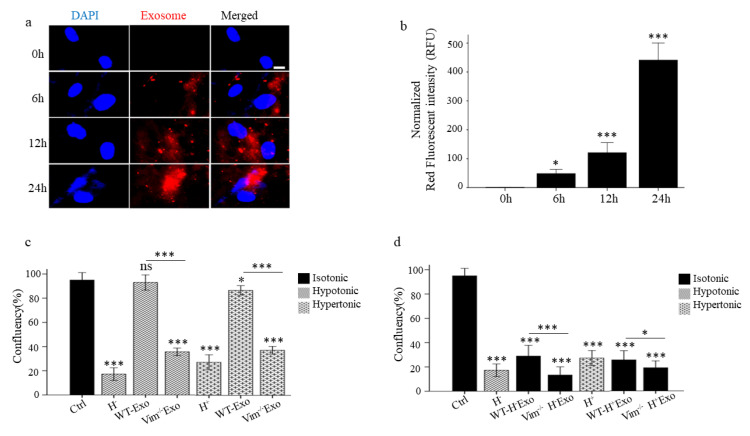
WT-APC-Exos promote osmotic stressed HDFs proliferation while exosomes from osmotic-stressed APCs decrease normal HDFs proliferation. (**a**) Representative images of exosome uptake by HDFs during 24 h treatment with WT-APC-Exo (100 μg/mL) followed by confocal microscopic observations. Scale bar = 5 μm. (**b**) RFU quantification of uptake WT-APC-Exos by HDFs. Data were normalized to surface area and represented as mean ± standard error of the mean (*n* = 10). * *p* < 0.05, *** *p* < 0.01. (**c**) Confluency percentage of osmotic-stressed HDFs treated with WT and Vim−/−Exos. (**d**) Confluency percentage of HDFs treated with hypo and hyper-stressed Exos (WT-H + Exos, Vim−/−H + Exos, WT-H + Exos, and Vim−/−H + Exos), respectively. Confluency percentage measured after 48 h using IncuCyte™ S3. Scale bar 100 μm. * *p* < 0.05, *** *p* < 0.001, ns: non-significant.

**Figure 4 ijms-22-04678-f004:**
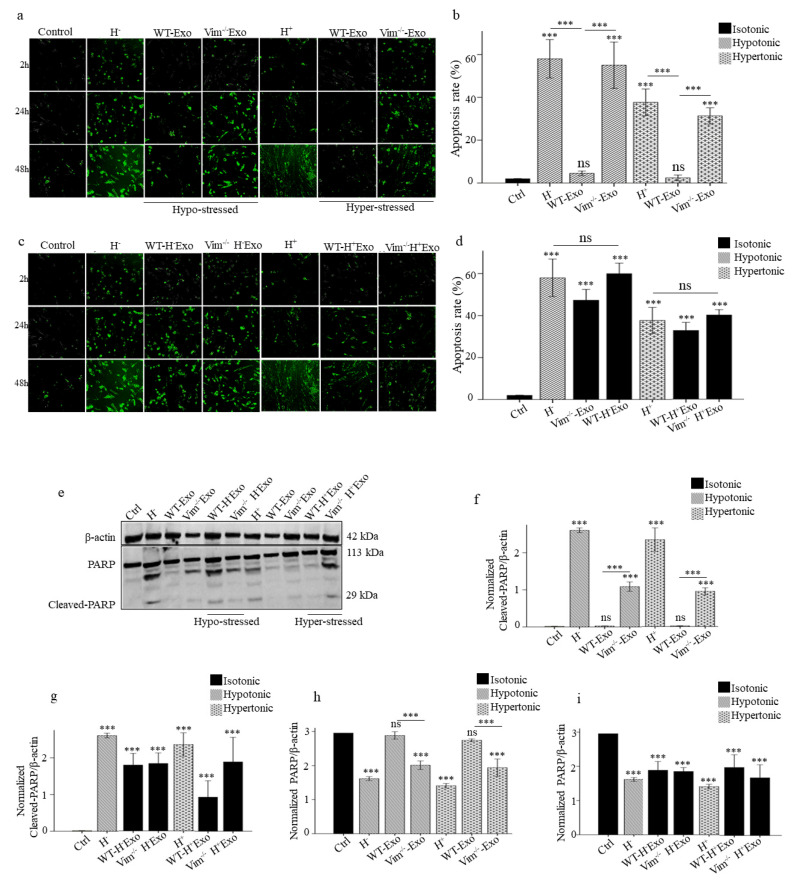
WT-APC-Exos prevent HDFs apoptosis induced by osmotic stress while exosomes from osmotic-stressed APCs induced normal HDFs apoptosis. (**a**) Representative images of activated caspase-3/7 and (**b**) apoptosis rate of osmotic-stressed HDFs treated with WT and Vim−/−Exos. (**c**) Representative images of activated caspase-3/7 and (**d**) apoptosis rate of HDFs treated with hypo and hyper-stressed Exos (WT-H + Exos, Vim−/−H + Exos, WT-H + Exos, and Vim−/−H + Exos). Apoptosis quantification is determined by an acquired fluorescent signal after 48h using an integrated object counting algorithm with IncuCyte™ S3, normalized against cell confluence. (**e**) Western blots with the anti-PARP antibody. β-Actin is used as a normalization control for the total lysate. The data shown were representatives of three independent experiments with similar results. (**f**) and (**g**) Pixel density quantification of cleaved-PARP and (**h**) and (**i**) PARP proteins from 3 independent experiments. Scale bar 100 μm. *** *p* < 0.001, ns: non-significant.

**Figure 5 ijms-22-04678-f005:**
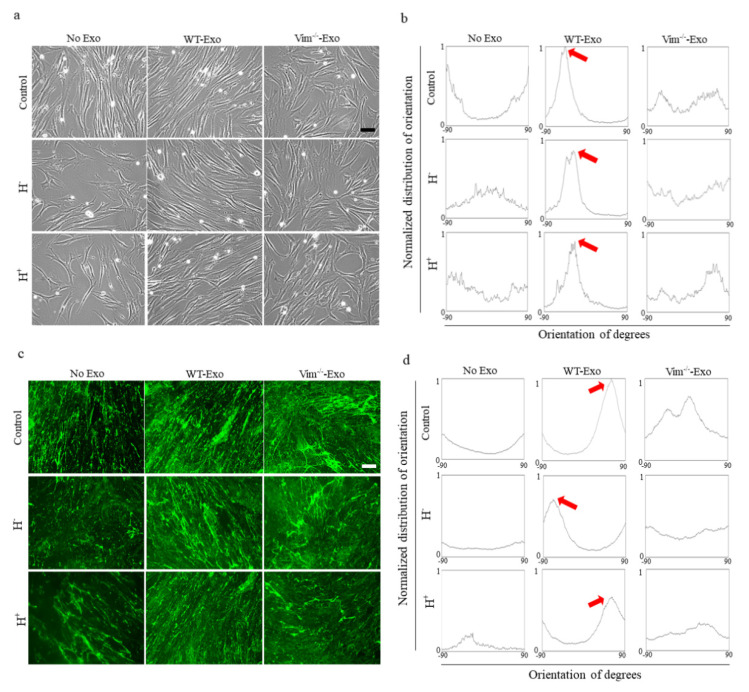
WT-APC-Exos affect collagen fiber orientation and promote ECM production by osmotic stressed HDFs. (**a**) Representative images and (**b**) computed directionality histogram of the HDF cell orientation treated with WT and Vim−/−Exos. (**c**) Representative images of the collagen I expression and (**d**) computed directionality histogram of collagen I fibers orientation in CDMs from HDFs treated with WT and Vim−/−Exos. (**e**) Relative fluorescent intensity quantification of the collagen I expression in CDMs from HDFs treated with WT and Vim−/−Exos. Scale bars: 50 μm. Data were normalized to surface area and represented as mean ± standard error of the mean (*n* = 5). Red arrows show dominant orientations. (**f**) Color maps of the collagen I fibers generated using the OrientationJ plugin. A particular orientation angle of the collagen is assigned to a color and the saturation of that color is assigned to the local coherency of the image.* *p* < 0.05, ** *p* < 0.01, *** *p* < 0.001.

**Figure 6 ijms-22-04678-f006:**
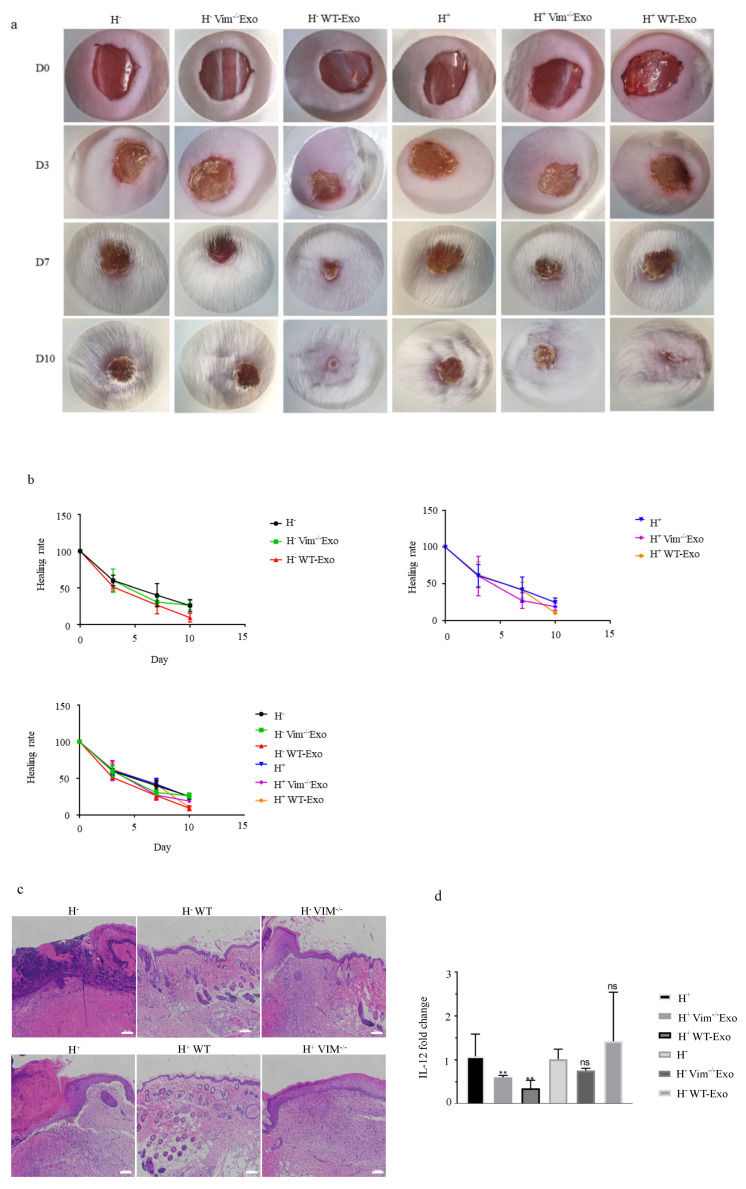
WT-APCs-Exos promote osmotic-stressed mouse model wound healing in vivo. (**a**) Macroscopic images of wounds from mice treated with WT-Exos (5 mg/kg) and Vim−/− Exos (5 mg/kg). Mice treated with hypo and hyper-osmotic solutions were used as controls. (**b**) Quantification of the speed of wound closure over 10 days (*n* = 3). (**c**) Hematoxylin and eosin (H&E) staining of wound tissue sections from each group of mice. (**d**) Quantification of the relative ratio of IL-12 in each group (*n* = 3). Scale bars: 100 μm. ** *p* < 0.01, ns: non-significant.

**Table 1 ijms-22-04678-t001:** List of primers for qPCR Analysis.

Gene	Forward Primer Sequence 5′→3′	Reverse Primer Sequence 5′→3′
Mouse-β-Actin	GGCTGTATTCCCCTCCATCG	CCAGTTGGTAACAATGCCATGT
Mouse-TNF-α	TCTCATCAGTTCTATGGCCC	GGGAGTAGACAAGGTACAAC
Mouse-Granzyme B	TCGACCCTACATGGCCTTAC	TGGGGAATGCATTTTACCAT
Mouse-IL-12	CAGCATGTGTCAATCACGCTAC	TGTGGTCTTCAGCAGGTTTC

## Data Availability

The authors confirm that the data supporting the findings of this study are available within the article and its Appendix A.

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
