# Peer review of "Exosomal Vimentin from Adipocyte Progenitors Protects Fibroblasts against Osmotic Stress and Inhibits Apoptosis to Enhance Wound Healing"

_ijms, 2021, doi:10.3390/ijms22094678_

Round 1

Reviewer 1 Report

Dear authors, I find your work well grounded and although in some parts there are some confusing sentences I think that your work contributes to the understanding of environmental queues on wound healing and the role of extra cellular vesicles in wound healing. I have included my comments in the attached document. I wish you all success.

Author Response

Answers to reviewers:

Dear reviewer,

Thanks for your valuable comments. We now changed the related parts in the text according to your comments. Since you put your comments in the original draft, we answered your comments in the same way. Furthermore, all your comments have been transferred and answered here as well.

Please be notified that since we made other changes according to the other reviewers as well, there have been some changes in the line and page numbers. We have mentioned your comment according to the original draft here and we addressed our answers according to the revised one.

1- Explain APCs. Page 2, line 87

Response from the authors: Thanks for your valuable comment. APCs were explained in line 84.

2- What does this elongation mean. I know this is "results", but I think it's important to make clear for any reader what the biological outcomes are. Page 3, line 100

Response from the authors: Thanks for your valuable comment. The following explanation was added to the text on page 3 line 99: “Since osmotic stress can affect cell volume, we measured these changes in the cell volume along a particular axis as a percentage of its original length and we represented it here as cell elongation.”.

3- Why did you do this ( assessed the expression levels of PARP and its cleaved form)? Again even if you and I might be experts and know why, it's important to explain your thought behind your choice of analysis. Page 7, line 179

Response from the authors: Thanks for your valuable comment. The following explanation was added to the text on page 7, line 189: “Poly(ADP-ribose) polymerase-1 (PARP-1) is a nuclear enzyme that is involved in the cellular response to DNA damage. Once PARP is cleaved by caspase during apoptosis, its DNA repair function is impaired.”

4- CDM, please spell out the abbreviation

Response from the authors: Thanks for your valuable comment. CDM (cell-derived matrices) has been explained first time in the method and material section on page 17, line 510.

5- This part of the sentence is a bit confusing. Maybe it can be rephrased (Page 9- line 228)

Response from the authors: Thanks for your valuable comment. Page 9, line 240 was replaced by the following sentence: “To study the collagen fiber orientation, we used CDMs secreted from osmotic-stressed and normal HDFs, where HDFs were treated with WT-Exos or Vim-/-Exos.”

6-what is "the constrained direction"? page 9, line 234

Response from the authors: Thanks for your valuable comment. Constrained here means in a certain or controlled direction. This word was replaced with the word “certain” in the text (page 9, line 246).

7- This part is a bit confusing to read. Please revise it and re-write this. Page 9 line 236

Response from the authors: Thanks for your valuable comment. This part (page 9, line 247) was rephrased accordingly: “Interestingly, there was a preferred fiber direction in HDFs treated with WT-Exos and fiber alignment was much more homogenous in this group compared to non-treated HDFs and HDFs treated with Vim-/-Exos (Fig 5(c) and (d)). Figures 5(b) and (d) show the distribution of the fibers’ orientation while dominant orientation is pointed out with the red arrows. According to the color-coded bar in Fig 5(f), fibers in WT- Exos samples appear in the same color (same direction) while no preferred fiber alignment was observed in the controls and Vim-/-Exos samples.”

8- This part as well needs to be re-wtitten using a more explanatory language. Page 9 line 245

Response from the authors: Thanks for your valuable comment. This part (page 9, line 256) was rephrased accordingly: In short, these results showed that the directionality of collagen fibers is similar to the directionality of the original cells, and compared to other treatments, WT-Exos can guide cells mainly to be oriented at the same angle.

9- cytoskeleton protein instead of cytoskeleton, page 14, line 339

Response from the authors: Thanks for your valuable comment. It was corrected in the text accordingly. Page 14, line 348

10- could be released should replace with could be released, page 14,line 339

Response from the authors: Thanks for your valuable comment. It was corrected accordingly. Page 14, line 348

11- Please specify in more detail what “using removal creams” means. Page 17, line 532:

Response from the authors: Thanks for your valuable comment. Following sentences are added to page 17, line 542: “To access wound healing, the dorsal dermal hair was removed by using the depilatory cream (VEET). Mice were injected with anesthetic by intraperitoneal injection, and then the mice went into a coma. The electric hairdresser (AUX-A8) was firstly used to remove part of the mice’ s dermal hair, after that, the depilatory cream (VEET) was evenly applied to the surgical site in a thin layer for 5 minutes, and then cotton balls which dipped in distilled water were used to gently wiped on the depilatory cream to remove both depilatory cream and the dorsal dermal hair”.

Reviewer 2 Report

I have some concerns and suggestions.

  1. In the Abstract, first row, insert 1-2 sentences referring material and methods used.
  2. Introduction and Results: specify what the acronyms signify WT-APCs, WT-Exo, Vim-/-APCs, Vim-/-Exo, HDF, WT-APC-Exos, WT-H- Exos, Vim-/-H- Exos, WT-H+Exos, Vim-/- H+ Exos, and so on). Too many and too complex acronyms for a not expert reader. The readers will surely find a lot of difficulty in understanding what has been done and what has been found. You must help them!
  3. Pag. 11, row 269: “to this aim” instead of “to this end”
  4. Pag. 13, row 328: correct “their” with “the”; row 331: …”WT-Exos involved in stress…”, insert “were” between involved and in
  5. Pag. 14, row 339: correct “which could release into” with “could be released”; row 343: "cellar"? Perhaps “cellular”
  6. The first and second sentences in the Conclusions section are hard to be understood. Re-write them trying to gain clarity.
  7. Throughout the text there are several grammar mistakes, see: inhibit instead of inhibits, compare instead of compared, etc. There is the need of an accurate revision.
  8. Fig 6c: for each group of micrographs there are two images showing different pictures. They are not labelled and the legend does not give information. Describe them accurately.

Author Response

Dear reviewer,

Thanks for your valuable comments. We now changed the related parts in the text according to your comments.

1- In the Abstract, first row, insert 1-2 sentences referring material and methods used.

Response from the authors: Thanks for your valuable comment. This sentence has been added to the abstract: “Here we performed in vitro and in vivo experiments to explore the effect of wide-type and vimentin knockout exosomes in accelerating wound healing under osmotic stress condition.” Page 1, line 19

2- Introduction and Results: specify what the acronyms signify WT-APCs, WT-Exo, Vim-/-APCs, Vim-/-Exo, HDF, WT-APC-Exos, WT-H- Exos, Vim-/-H- Exos, WT-H+Exos, Vim-/- H+ Exos, and so on). Too many and too complex acronyms for a not expert reader. The readers will surely find a lot of difficulty in understanding what has been done and what has been found. You must help them!

Response from the authors: Thanks for your valuable comment. All the abbreviations are explained in the text now.

Adipocyte progenitors are marked with (APCs) on page 2, line 83. Also, extra explanations were added on page 3, line  99 - 101 for (WT), (Vim-/-), (H-), and (H+)  accordingly.

Following text is also added on page 4, line 147 “For future clarification, we used following abbreviations for different isolated exosomes: exosomes from wild type adipocytes (WT-Exo), exosomes from vimentin knockout adipocytes (Vim-/-Exo), exosomes from wild type adipocytes under hypo-osmotic stress (WT-H-Exo), exosomes from vimentin knockout adipocytes under hypo-osmotic stress (Vim-/-H-Exo), exosomes from wild type adipocytes under hyper-osmotic stress (WT-H+Exo) and exosomes from vimentin knockout adipocytes under hyper-osmotic stress (Vim-/-H+Exo).”

3- Pag. 11, row 269: “to this aim” instead of “to this end”

Response from the authors: Thanks for your valuable comment. It was corrected accordingly. Page 11,line 278

4- Pag. 13, row 328: correct “their” with “the”; row 331: …”WT-Exos involved in stress…”, insert “were” between involved and in

      Response from the authors: Thanks for your valuable comment. It was corrected accordingly. Page 13, line 337 and 340

5- Pag. 14, row 339: correct “which could release into” with “could be released”; row 343: "cellar"? Perhaps “cellular”

Response from the authors: Thanks for your valuable comment. It was corrected accordingly. Page 14, line 348 and 352

6- The first and second sentences in the Conclusions section are hard to be understood. Re-write them trying to gain clarity.

Response from the authors: Thanks for your valuable comments. The following sentences were replaced: “Altogether, the results of this study for the first time indicated that exosomes can be considered as a complex information package to either modify and restore the osmotic balance or to convey and induce osmotic stress-driven condition, while exosomal vimentin significantly contributed to this process. However, the mechanisms by which exosomes might acquire new biological functions in the stress microenvironment remains to be determined.”. Page 18, line 585

7- Throughout the text there are several grammar mistakes, see: inhibit instead of inhibitscompare instead of compared, etc. There is the need of an accurate revision.

Response from the authors: Thanks for your valuable comment. About the word “inhibit”, could you please specify those sentences. We were considering exosomal vimentin and osmotic stress in singular format. Otherwise, we used this word in plural form. About the word “compare”, we have changed it through the text.

8- Fig 6c: for each group of micrographs there are two images showing different pictures. They are not labelled and the legend does not give information. Describe them accurately.

Response from the authors: Thanks for your valuable comment. We have labeled the imagines and we have described them.

Round 2

Reviewer 2 Report

I have two more concerns:

Pag. 3, row 100. Are you sure to maintain "such"? I believe it has to be deleted

Pag. 18, row 595. Conclusions.  Re-write the sentence: "However the mechanisms which exosomes might acquire new biological functions in the stress microenvironment remains to be determined". Its significance is obscure: However it remains to be determined What??? The mechanisms ???? which ???? exosomes might acquire ??? new biological functions ????? in the stress microenvironment ?????

Author Response

Dear reviewer,

Thanks for your valuable comments. We now changed the related parts in the text according to your comments.

1- Page 3, row 100. Are you sure to maintain "such"? I believe it has to be deleted

Response from the authors: Thanks for your valuable comment. The word “such” was deleted.

2- Page 18, row 595. Conclusions.  Re-write the sentence: "However the mechanisms which exosomes might acquire new biological functions in the stress microenvironment remains to be determined". Its significance is obscure: However it remains to be determined What??? The mechanisms ???? which ???? exosomes might acquire ??? new biological functions ????? in the stress microenvironment ?????

Response from the authors: Thanks for your valuable comment. Now we combined two sentences in one as following: “However, more mechanistic studies are needed to illuminate how stress conditions affect exosome-mediated intercellular communication, signaling pathways, and phenotypic behavior of recipient cells.”
